# Transcriptomic, Physiological, and Metabolomic Response of an Alpine Plant, *Rhododendron delavayi*, to Waterlogging Stress and Post-Waterlogging Recovery

**DOI:** 10.3390/ijms241310509

**Published:** 2023-06-22

**Authors:** Xi-Min Zhang, Sheng-Guang Duan, Ying Xia, Jie-Ting Li, Lun-Xian Liu, Ming Tang, Jing Tang, Wei Sun, Yin Yi

**Affiliations:** 1Key Laboratory of Plant Physiology and Development Regulation, Guizhou Normal University, Guiyang 550025, Chinallunxian@163.com (L.-X.L.); tangjing2016@gznu.edu.cn (J.T.); sunwei889@163.com (W.S.); gzklppdr@gznu.edu.cn (Y.Y.); 2Key Laboratory of Environment Friendly Management on Alpine Rhododendron Diseases and Pests of Institutions of Higher Learning in Guizhou Province, Guizhou Normal University, Guiyang 550025, China; 3School of Life Sciences, Guizhou Normal University, Guiyang 550025, Chinamingtang@gznu.edu.cn (M.T.); 4Key Laboratory of State Forestry Administration on Biodiversity Conservation in Karst Area of Southwest, Guizhou Normal University, Guiyang 550025, China

**Keywords:** *Rhododendron delavayi*, waterlogging, photosynthesis, lignin, hydrogen peroxide, soluble sugar

## Abstract

Climate change has resulted in frequent heavy and prolonged rainfall events that exacerbate waterlogging stress, leading to the death of certain alpine *Rhododendron* trees. To shed light on the physiological and molecular mechanisms behind waterlogging stress in woody *Rhododendron* trees, we conducted a study of *Rhododendron delavayi*, a well-known alpine flower species. Specifically, we investigated the physiological and molecular changes that occurred in leaves of *R. delavayi* subjected to 30 days of waterlogging stress (WS30d), as well as subsequent post-waterlogging recovery period of 10 days (WS30d-R10d). Our findings reveal that waterlogging stress causes a significant reduction in CO_2_ assimilation rate, stomatal conductance, transpiration rate, and maximum photochemical efficiency of PSII (Fv/Fm) in the WS30d leaves, by 91.2%, 95.3%, 93.3%, and 8.4%, respectively, when compared to the control leaves. Furthermore, the chlorophyll a and total chlorophyll content in the WS30d leaves decreased by 13.5% and 16.6%, respectively. Both WS30d and WS30d-R10d leaves exhibited excessive H_2_O_2_ accumulation, with a corresponding decrease in lignin content in the WS30d-R10d leaves. At the molecular level, purine metabolism, glutathione metabolism, photosynthesis, and photosynthesis-antenna protein pathways were found to be primarily involved in WS30d leaves, whereas phenylpropanoid biosynthesis, fatty acid metabolism, fatty acid biosynthesis, fatty acid elongation, and cutin, suberin, and wax biosynthesis pathways were significantly enriched in WS30d-R10d leaves. Additionally, both WS30d and WS30d-R10d leaves displayed a build-up of sugars. Overall, our integrated transcriptomic, physiological, and metabolomic analysis demonstrated that *R. delavayi* is susceptible to waterlogging stress, which causes irreversible detrimental effects on both its physiological and molecular aspects, hence compromising the tree’s ability to fully recover, even under normal growth conditions.

## 1. Introduction

Climate change has had a profound impact on the global hydrological cycle, resulting in a marked increase in heavy or recurrent precipitation events in certain regions. Such rainfall frequently exceeds the soil’s water-holding capacity, resulting in oversaturation of terrestrial ecosystems, especially in zones with poor drainage, clay-rich soils, and low-lying areas. This oversaturation can trigger severe waterlogging stress as a result of soil ponding [1,2,3].

Waterlogging stress impedes the gas exchange between plant tissues and soil, causing anoxic or hypoxic conditions in the waterlogged tissues. These conditions can cause an “energy crisis” due to the inhibition of ATP generation [1,4]. Plant responses to waterlogging stress involve complex signal transduction pathways that alter the synthesis and balance of plant hormones, resulting in stomatal closure and morphological changes [5,6]. Additionally, waterlogging stress induces the accumulation of plant metabolites such as flavonoids, vitamin B6, and sugars that play a crucial role in the response to waterlogging stress [7]. Previous studies have shown that hormone mediated transcription factors, including MYBs, ERFs, WRKYs, and NACs, regulate plant responses to waterlogging stress [8,9]. Moreover, waterlogging triggers anaerobic metabolism in roots, leading to the production of ethanol, acetaldehyde, and other harmful substances that can poison root cells and damage protein structure [10,11]. Ultimately, this leads to accelerated root decay, damage, and even death [12].

Currently, extensive research has been conducted on the molecular and physiological responses to waterlogging stress in model plants such as Arabidopsis and crops such as *Solanum lycopersicum*, *Cucumis sativus*, and *Glycine max* [2,13]. However, trees play a crucial ecological role in global climate change and are essential for urban environments, such as greening streets and parks. Therefore, it is crucial to understand the mechanisms of underlying tree response to waterlogging stress. Although some studies have reported on the physiological response mechanisms of waterlogging-sensitive species or varieties [3,14,15,16], only a few have investigated changes in the transcriptome and metabolome of woody economic species, such as poplar, cotton, oak, kiwifruit, and *Guazuma ulmifolia*, exposed to waterlogging stress [17,18,19,20]. Thus, there is a need to investigate the response mechanisms of specific woody trees to waterlogging stress, as given that their tolerance mainly depends on the species and degree of waterlogging [21].

*Rhododendron*, the largest genus in the Ericaceae family and one of the most notable alpine flowers, comprises approximately 1143 species of evergreen and deciduous woody plants. In the northwest Guizhou province of China, wild *Rhododendron* trees in the Baili Azalea Nature Reserve (BANR) represent significant potential tourism resources for local communities. Although these trees typically grow in high mountain areas, some low-lying areas or clay-rich soils have experienced long-term ponding in recent years, hindering the growth of *Rhododendron* species and leading to the death of some trees. Despite the importance of Rhododenron trees in this region, there is currently a scarcity of research on the physiological and molecular response to waterlogging stress and post-waterlogging recovery in this genus.

*Rhododendron delavayi*, a crucial member of the *Rhododendron* genus in the BANR, has unfortunately experienced mortality in low-lying areas or clay-rich soils due to continuous rainfall. This has prompted us to urgently investigate the response mechanisms of *R. delavayi* to waterlogging stress. In this study, *R. delavayi*, an alpine plant, has been utilized as a model system to explore the physiological and molecular responses to waterlogging stress and post-waterlogging recovery. This approach aims to provide key insights into how alpine plants respond to waterlogging stress both at physiological and molecular levels. Ultimately, our findings can contribute a fundamental understanding of the death of *R. delavayi* caused by waterlogging stress.

## 2. Results

### 2.1. Waterlogging Stress Caused the Aged Leaf Wilting in R. delavayi Seedlings

To explore the physiological and molecular responses of *R. delavayi* to waterlogging stress, *R. delavayi* seedlings were subjected to various durations of waterlogging stress (WS10d, WS20d, and WS30d) followed by a post-waterlogging recovery period (WS10d-R10d, WS20d-R10d, and WS30d-R10d). During WS10d, WS20d, WS10d-R10d, and WS20d-R10d, no stress symptoms were observed when compared to their pre-waterlogging state (Figure 1). However, prolonged waterlogging stress (WS30d) led to the wilting of aged leaves in some seedlings (as indicated by white arrow in Figure 1) and even 10 days of post-waterlogging recovery was insufficient to fully restore the wilted leaves to their pre-waterlogging state (as indicated by red arrow in Figure 1). These results suggested that waterlogging stress lasting for 30 days can severely damage *R. delavayi* seedlings.

### 2.2. Transcriptome Assembly, Analysis of Differentially Expressed Genes (DEGs), and qRT-PCR Validation

To investigate the molecular mechanisms underlying the response of *R. delavayi* to waterlogging stress at the transcriptome level, RNA sequencing was performed on leaves from CK, WS30d, and WS30d-R10d plants, based on the stress symptoms of *R. delavayi* leaves caused by waterlogging stress. Over 20,163,917 clean reads were obtained from each sample, with the lowest mapped ratio being 76.95% in WS30d-R10d-2. The guanine and cytosine/total base (GC content) and the quality of base calling accuracy at 99.9% (Q30) were more than 47.42% and 92.33%, respectively (Appendix A). After assembly of the high-quality reads, a total of 97,152 unigenes and 402,244 trancripts with mean lengths of 870 bp and 1705 bp, respectively, were obtained (Appendix A).

To determine the variation and similarity of the gene expression profiles among the samples, we conducted principal component analysis (PCA) of all detected genes using normalized Fragments Per Kilobase of transcript per Million mapped reads (FPKM) values. The difference in gene expression among the three samples was statistically significant (*p* = 0.032), with PC1 and PC2 accounting for 58.5% and 26.4% of the total variation, respectively (Figure 2A). The PCA plot showed that the data from the three biological replicates were closely clustered together and separated between CK and WS30d and WS30d-R10d (Figure 2A), demonstrating that the sequencing data from stressed leaves differed significantly from those of the CK leaves.

We subsequently identified differentially expressed genes (DEGs) among the WS30d, WS30d-R10d, and CK samples. Comparion of WS30d leaves to the CK leaves revealed 2521 up-regulated and 2774 down-regulated DEGs, while comparison of WS30d-R10d to CK leaves identified 1589 up-regulated and 1840 down-regulated DEGs (Figure 2B, Appendix A). DEGs between WS30d and WS30d-R10d were also obtained (Appendix A). These results suggested that waterlogging stress and post-waterlogging recovery significantly induced gene transcription in *R. delavayi* leaves.

To verify the reliability of DEGs identified by RNA sequencing, we randomly selected six genes and quantified their expression levels using quantitative Real-Time PCR (qRT-PCR) analysis. The results demonstrated the consistency between the qRT-PCR data and the RNA sequencing data, thereby confirming the reliability of the DEG screened by RNA sequencing (Appendix A).

### 2.3. KEGG Pathways by DEGs of Waterlogging Stress Versus Control

To further comprehend the biological functions of the 5295 DEGs induced by 30 days of waterlogging stress, we performed Kyoto Encyclopedia of Genes and Genomes (KEGG) pathway analysis. The up-regulated DEGs were significantly enriched in the “purine metabolism” and “glutathione metabolism” pathways (*p* < 0.05) (Figure 3A), while the down-regulated DEGs were significantly enriched in “photosynthesis”, “photosynthesis-antenna proteins”, “fat acid biosynthesis”, and “glycosaminoglycan degradation” pathways (*p* < 0.05) (Figure 3B).

In the “purine metabolism” pathway, the enzymes encoded by the up-regulated DEGs directly or indirectly catalyzed ATP production (Table 1), indicating their crucial role in providing energy. In the “Glutathione metabolism” pathway, 21 up-regulated DEGs were annotated as glutamate-cysteine ligase, glutathione peroxidase, glutathione reductase, and isocitrate dehydrogenase, which participate in the glutathione cycle (Table 1) and facilitate the accumulation of glutathione, as evidenced by metabolome detection (Table 2). Furthermore, four DEGs were up-regulated and annotated as L-ascorbate peroxidase and ribonucleoside-diphosphate reductase (Table 1), suggesting their involvement in the ascorbate cycle.

In the “photosynthesis” pathway, the down-regulated DEGs were annotated to encode proteins such as cytochrome b6/f complex, electron transport chain components, ATPase, and subunits of photosystem I (PSI) or II (PSII) (as shown in Table 1). Further analysis revealed a significant reduction in CO_2_ assimilation rate, stomatal conductance, transpiration rate, and maximum photochemical efficiency by 91.2%, 95.3%, 93.3%, and 8.4%, respectively, in the WS30d leave in comparison with CK (Figure 4). Additionally, the chlorophyll a and total chlorophyll content in the WS30d leaves decreased by 13.5% and 16.6%, respectively (Figure 4). These observations suggested that excessive light energy might lead to the accumulation of reactive oxygen species (ROS) in the chloroplast. This was further supported by the significant increase in hydrogen peroxide content in WS30d leaves, which was found to be 3.39 times higher than the control (Figure 4).

In the “fatty acid biosynthesis” pathway, down-regulated DEGs were annotated as FabF, FabH, FabI, fatty acyl-ACP thioesterase, and long-chain acyl-CoA synthetase (Table 1). These genes are involved in synthesizing of long-chain acyl-CoA. Similarly, in the “glycosaminoglycan degradation” pathway, down-regulated DEGs were annotated as heparanase and hexosaminidase (Table 1).

### 2.4. KEGG Pathways by Common DEGs between Waterlogging Stress and Post-Waterlogging Recovery

Our objective was to identify genes that are difficult to recover after waterlogging stress. To accomplish this, we analyzed the common DEGs between WS30d and WS30d-R10d using a Venn diagram, which revealed 2417 common DEGs (Figure 5A). Heat map analysis of these DEGs indicated that 1037 DEGs were up-regulated, and 1380 DEGs were down-regulated (Figure 5B). Subsequently, we examined the biological relevance of these DEGs by using KEGG pathway analysis. The up-regulated DEGs were significantly enriched in “selenocompound metabolism” (*p* < 0.05) (Figure 5C), whereas the down-regulated DEGs were significantly enriched in “phenylpropanoid biosynthesis”, “fatty acid metabolism”, “fatty acid biosynthesis”, “fatty acid elongation”, and “cutin, suberin, and wax biosynthesis” (*p* < 0.05) (Figure 5D).

In the “selenocompound metabolism” pathway, we observed up-regulated DEGs annotated as 3′-phosphoadenosine 5′-phosphosulfate synthase and methionyl-tRNA synthase (Appendix A). Conversely, in the “phenylpropanoid biosynthesis” pathway, we identified down-regulated DEGs annotated as 4-coumarate-CoA ligase, beta-glucosidase, caffeic acid 3-O-methyltransferase, cinnamyl-alcohol dehydrogenase, coumaroylquinate 3′-monooxygenase, peroxidase, and shikimate O-hydroxycinnamoyltransferase (Appendix A). These genes are mainly involved in the biosynthesis of lignin.As expected, the lignin content of WS30d-R10d leaves decreased significantly when compared to CK (Figure 6).

In the “fatty acid metabolism” and “fatty acid biosynthesis” pathways, the proteins encoded by DEGs were consistent with those in the “fatty acid biosynthesis” pathway mentioned above (Appendix A). The down-regulated DEGs may impede the synthesis of long-chain acyl-CaA. In the “fatty acid elongation” pathway, DEGs were annotated as 3-ketoacyl-CoA synthase, very-long-chain (3R)-3-hydroxyacyl-CoA dehydratase, and very-long-chain 3-oxoacyl-CoA reductase (Appendix A). The decreased expression of these DEGs suggested that the synthesis of long-chain fatty acids with long-chain acyl-CaA as a precursor may be inhibited. Long-chain fatty acids are utilized for the biosynthesis of cutin, suberin, and wax. The down-regulated DEGs involved in these pathways may impede the biosynthesis of long-chain esters (Appendix A).

### 2.5. Major Transcription Factors Families Active during Waterlogging Stress and Post-Waterlogging Recovery

Transcription factors (TFs) play a crucial role in the transcriptional reprogramming that occurs in response to abiotic stress. To identify the enriched TF families in response to waterlogging stress, we screened the DEGs in WS30d leaves. Our analysis revealed that 20 TF families were encoded by 118 up-regulated DEGs and 117 down-regulated DEGs (Figure 7). Among the up-regulated DEGs, the WRKY family was the most dominant, while the bHLH family was the most dominant among the down-regulated DEGs. Interestingly, the DEGs encoding most TF families, such as MYB and AP2/ERF-ERF, were both up-regulated and down-regulated under waterlogging stress (Figure 7). In the common DEGs between WS30d and WS30d-R10d, the MYB-related and bHLH families were the most dominant TF families encoded by the up-regulated and down-regulated DEGs, respectively (Appendix A).

### 2.6. Metabolites Accumulations in Response to Waterlogging Stress and Post-Waterlogging Recovery

To investigate the accumulation of metabolites in response to waterlogging stress and post-waterlogging recovery, we utilized the GC-MS platform to determine the metabolite levels in CK, WS30d, and WS30d-R10d leaves. To summarize the similarities and differences between these groups, we employed principal component analysis (PCA) and orthogonal projections to latent structures-discriminant analysis (OPLS-DA). Our PCA analysis revealed that the data from the three duplicate samples were closely clustered, with PC1 and PC2 accounting for 20.6% and 16.7% of the total variation, respectively (Appendix A). In the OPLS-DA score plot, component 1 and component 2 accounted for 19.8% and 12.2% of the total variation, respectively (Appendix A). We identified a total of 149 putative metabolites and screened for differential metabolites between treatment groups (VIP > 1, and *p* < 0.05). Compared to CK, WS30d leaves showed significant upregulation in 22 metabolites, including seven sugars (glucose, sedoheptulose, galactose, sucrose, lyxose, galactonic acid, N-Acetyl-beta-D-mannosamine), three organic acids, three lipids, two alcohols, two flavonoids, and three others (Table 2). WS30d-R10d leaves showed upregulation in six sugars, three lipids, two alcohols and flavonoids, and five others (Appendix A). Interestingly, the common metabolites in WS30d and WS30d-R10d leaves revealed that soluble sugars remained highly accumulated during post-waterlogging recovery (Appendix A).

## 3. Discussion

### 3.1. Waterlogging Stress Inhibited Photosynthesis in R. delavayi Leaves

Gas exchange is usually used to assess the tolerance of plant species to waterlogging [21]. Tolerant species show a slight reduction in their photosynthetic rate [3], whereas sensitive species experience a significant reduction [22,23]. In the case of *R. delavayi*, the CO_2_ assimilation rate in WS30d leaves decreased by 91.2%, suggesting that it is a waterlogging-sensitive species (Figure 4A), similar to *Quercus petraea* and *Persea Americana* [23,24]. This decrease in CO_2_ assimilation rate in WS30d leaves was accompanied by a simultaneous reduction in stomatal conductance and transpiration rate (Figure 4B,C), indicating that photosynthesis may be affected by stomatal limitation [25]. Although the precise mechanism behind the decrease in stomatal conductance caused by waterlogging stress remains unclear, it has been widely reported. In addition, the decrease in maximum photochemical efficiency of PSII (Fv/Fm) in WS30d leaves (Figure 4D) suggested that photoinhibition had occurred in the stressed leaves [26,27].

The suppression of photosynthesis could also be affected by both chlorophyll content and photosynthetic enzyme activity. In the case of WS30d leaves, there was a decrease of 13.5% and 16.6% in chlorophyll a and total chlorophyll content, respectively (Figure 4), which is consistent with previous studies [3,27]. Additionally, the decrease in chlorophyll a and chlorophyll (a + b) content in stressed leaves indicates that the inhibition of photosynthesis, which results in more damage to PSII, is also more limiting [3]. This, in turn, may lead to the production of excessive ROS. RNA-sequencing analysis of stressed leaves revealed that down-regulated DEGs were enriched in “photosynthesis” and “photosynthesis-antenna proteins” pathways (Figure 3B). The DEGs annotated in these pathways were found to be involved in PSI, PSII, cytochrome *b6-f* complex, and ATP synthase (Table 1), suggesting that the activity of proteins in the photosynthesis pathway was inhibited by waterlogging stress. Although the chlorophyll a and chlorophyll (a + b) content in WS30d-R10d leaves could be restored to the control level, the CO_2_ assimilation rate was still reduced (Figure 4). These findings are similar to previous studies on cotton, a woody plant that is sensitive to waterlogging stress [28,29].

Transcription factors play a crucial role in regulating gene expression, particularly in the synthesis of plant secondary metabolites such as anthocyanins, flavonoids, and lignin. MYB and bHLH transcription factors, in particular, have been identified as key regulators of this process [30,31]. In waterlogged leaves, it is posited that the down-regulation of MYB and bHLH genes may exert negative regulation on the expression of genes involved in flavonoid biosynthesis, leading to flavonoid accumulation, or alternatively, positively regulate genes involved in lignin biosynthesis, thereby inhibiting lignin synthesis (Figure 6). Further, WRKY and MYB-related transcription factors are instrumental in regulating ABA biosynthesis [30,32], a process that has been shown to play a significant role in the waterlogged plants [33]. It is hypothesized that the up-regulation of WRKY- and MYB-related transcription factors in waterlogged leaves may regulate ABA biosynthesis, ultimately causing leaf stomatal closure and reduction in stomatal conductance (Figure 4). Additionally, plant hormones induce the expression of ethylene response factors, such as *RAP2.3*, improving expression levels of downstream genes and enhancing waterlogging or post-waterlogging resistance [32,34]. Based on our findings, it is postulated that the up-regulation of ERF transcription factors may stimulate H_2_O_2_ accumulation in the stressed leaves by ethylene biosynthesis (Figure 4). Notably, the observed alterations in the expression levels of both up-regulated and down-regulated ERF family members in conjunction with the varying expression patterns observed in other transcription factors and plant hormone signaling pathways indicates a highly intricate regulatory network governing the plant response to waterlogging stress [17].

### 3.2. Waterlogging Stress Induced Oxidative Stress in R. delavayi Seedlings

The existing body of evidence indicates that short-term waterlogging stress has the potential to generate ROS in plant roots. In leaf tissues, several abiotic stressors can result in a reduction in CO_2_ assimilation rate, resulting in the accumulation of excess H_2_O_2_ and consequent oxidative stress [35]. In addition, waterlogging stress appears to alter the accumulation of ethylene and provoke the production of H_2_O_2_, as evidenced by the increased H_2_O_2_ accumulation in the WS30d leaves (Figure 4H). Such H_2_O_2_ accumulation may disrupt normal metabolic processes by promoting lipid peroxidation, compromising membrane integrity, as well as protein and DNA oxidation [36,37,38]. Analysis of gene expression patterns revealed that the up-regulated DEGs in the stressed leaves were significantly enriched in the “purine metabolism” pathways (Figure 3A), suggesting that the plants may initiate gene expression mechanisms to repair DNA damage caused by H_2_O_2_ accumulation (Table 1).

In the context of ROS accumulation under waterlogging stress, the ascorbate-glutathione cycle plays a critical role in enabling plants to cope with oxidative stress to a certain extent [39]. Notably, the observed increases in ascorbate or glutathione concentrations in the WS30d leaves or WS30d-R10d (Table 2 and Appendix A) were found to be consistent with the up-regulated DEGs in the glutathione metabolism pathway (Figure 3A). This strongly suggests that the accumulation of glutathione or ascorbate could serve as non-enzymatic antioxidants to preserve the balance of H_2_O_2_. However, despite the fact that waterlogged seedlings were able to recover, their H_2_O_2_ content remained at a higher level than the post-waterlogging recovery (Figure 4H). As such, oxidative stress was sustained and even accelerated the damage inflicted on the plants during the recovery, as shown in Figure 1.

### 3.3. Lignin and Cuticle Biosynthesis was Continuously Inhibited during Post-Waterlogging Recovery

Lignin, a phenolic biopolymer, is primarily synthesized through the phenylpropanoid biosynthesis pathway and is typically concentrated in the secondary cell wall of vascular plants. Lignin plays a vital role in providing mechanical support to plant tissues, as well as facilitating the transportation of nutrients or carbohydrates [40]. Notably, the accumulation of lignin is known to enhance cell wall reinforcement, thereby improving resistance to waterlogging stress [41]. Previous studies have shown that waterlogging stress can reduce lignin content by inhibiting the expression of genes involved in lignin biosynthesis [42]. Our transcriptome data analysis revealed that the DEGs in WS30d-R10d leaves were significantly enriched in the phenylpropanoid biosynthesis pathway (Figure 5D), with their expression being noticeably inhibited (Appendix A). These results were strongly aligned with the observed decrease in lignin content in WS30d-R10d leaves (Figure 6). As such, we hypothesize that waterlogging may reduce the biosynthesis of lignin in the stressed leaves, ultimately leading to leaf wilting (Figure 1) and decreased nutrient or carbohydrate transportation (Figure 8). A similar result was reported in poplar trees [17].

The cuticle is a layer of fatty substances that envelops above-ground plant organs, composed mainly of cutin and waxes [43]. As reported by Tellechea-Robles et al. (2019), wetland plants have adapted to waterlogging stress by augmenting the thickness of their cuticle on leaves [44]. In this study, transcriptome analysis revealed that the DEGs were significantly enriched in several pathways during post-waterlogging recovery, namely “fatty acid metabolism”, “fatty acid biosynthesis”, “fatty acid elongation”, and “cutin, suberin, and wax biosynthesis” (Figure 5D). These pathways may reduce the production of cutin and wax, leading to thinner leaves and a diminished support function.

### 3.4. Waterlogging Stress Prevented Transportation of Soluble Sugar from Leaves

Maintaining a stable supply of glycolysis, particularly in roots, is a crucial factor in waterlogging tolerance. In contrast, waterlogging-sensitive plants often fail to uphold sufficient carbohydrate levels, leading to tissue death under stress conditions [21]. Previous studies have established that waterlogging-sensitive species frequently experience substantial disruptions in sugar transportation from phloem to root cells during periods of waterlogging, resulting in increased sugar accumulation in stressed leaves [22,28,45,46]. Our current study confirmed that soluble sugars, such as glucose, sedoheptulose, galactose, and sucrose, were significantly accumulated in waterlogged leaves as compared to CK leaves (Table 2). In addition, we noted excessive accumulation of soluble sugar in the stressed leaves during post-waterlogging recovery (Appendix A). These findings suggest that sugar transport in *R. delavayi* seedlings may have been severely impaired by waterlogging stress, which ultimately resulted in a deficiency of available carbohydrates in the roots [28].

## 4. Materials and Methods

### 4.1. Plant Material and Waterlogging Treatment

Three-year-old *Rhododendron delavayi* seedlings were cultivated in plastic flowerpots (8 cm in diameter and 8 cm in height), filled with a mixture of nutrient soil and humus soil (1:1 in volume). To induce waterlogging stress, the seedlings were placed within a greenhouse (with a photoperiod 16 h/8 h, temperature 22 °C, light intensity 400 μmol m^−2^ s^−1^, relative humidity 60–70%) at Guizhou Normal University.

The flowerpots with the seedlings were placed in trays with length, width, and height of 55 cm, 30 cm, and 4.5 cm, respectively. The trays were filled with water up to a height of 4 cm, submerging the root area of the *R. delavayi* seedlings. The water level was sustained at 4 cm by daily additions, creating a waterlogging stress that was imposed for 10 days (WS10d), 20 days (WS20d), and 30 days (WS30d). After the waterlogging stress period, the water within the trays was removed, and the seedlings were allowed to recover for 10 days, marked as WS10d-R10d, WS20d-R10d, and WS30d-R10d, respectively. *R. delavayi* seedlings before the waterlogging stress were used as the control (CK). Three flowerpots were placed within each tray as a repeat, with three biological repeats set for each treatment.

### 4.2. Measurement of Photosynthesis Parameters and Chlorophyll Fluorescence

To measure photosynthesis parameters, we utilized the portable photosynthesis system (LI-6400, LI-COR Corporate, Lincoln, NE, USA), which was coupled with the 6400-02B chamber (6400-02B, LI-COR Corporate, Lincoln, NE, USA) providing stable and activated intensities. The third young leaf from three individuals was selected and measured between 8.00 and 12.00 a.m. Prior to measurement, the leaves were activated using a light intensity of 1000 µmol quanta m^−2^ s^−1^ for 20 s. Subsequently, the light intensity (400 µmol quanta m^−2^ s^−1^) and CO_2_ concentration (ambient CO_2_ concentration) were set, and the net CO_2_ assimilation rate, stomatal conductance, and transpiration rate were recorded. Additionally, we measured chlorophyll fluorescence using the portable photosynthesis system (LI-6400, LI-COR Corporate, Lincoln, NE, USA) with the 6400-40 chamber. After dark adaptation at night, we conducted measurements before dawn, recording the maximum quantum efficiency of photosystem II (Fv/Fm).

### 4.3. Measurement of Hydrogen Peroxide and Lignin

In this study, we extracted 0.5 g of leaves and subsequently homogenized them with 3 mL of 50 mmol/L phosphate buffer (pH = 6.5). Next, we centrifuged the mixture at 1000 rpm for 5 min at 4 °C, following which we added 1 mL of 0.1% titanium sulfate solution to 3 mL of supernatant and mixed it thoroughly. The solution was then centrifuged at 1000 rpm for 10 min at 4 °C. The resulting precipitate was washed with cooled acetone and subsequently dissolved in 2 mmol/L H_2_SO_4_ solution. Subsequently, we measured the absorbance at 410 nm and the hydrogen peroxide content was calculated using the standard curve. Meanwhile, lignin was extracted using an assay kit (BC4200, Solarbio Technology Co., Ltd., Beijing, China) and determined according to the instructions.

### 4.4. RNA Extraction, Library Preparation, and Sequencing

After a 30-day period of exposure to waterlogging stress, the aged leaves of the plant under investigation exhibited signs of wilting (Figure 1). To gain insight into the effects of this stress, we selected CK, WS30d, and WS30d-R10d leaves and roots for RNA sequencing (RNA-Seq). Total RNA was extracted from the samples using Trizol reagent (Thermo Fisher Scientific, Pleasanton, CA, USA) and evaluated for RNA degradation and contamination using agarose gelelectrophoresis (1%). RNA purity was checked using the Nano Photometer spectrophotometer (Thermo Fisher Scientific, Wilmington, NC, USA), and RNA concentration was measured using the Qubit RNA Assay Kit in Qubit 2.0Fluorometer (Thermo Fisher Scientific, Wilmington, NC, USA). RNA integrity was assessed using the RNA Nano6000 Assay Kit of the Agilent Bioanalyzer 2100 system (Agilent Technologies, Santa Clara, CA, USA). The NEBNext UltraRNA Library Prep Kit for Illumina (New England Biolabs, Inc., Beijing, China) was used to generate sequencing libraries according to the manufacturer’s recommendations. However, due to the short roots and lignification of *R. delavayi* seedlings, RNA extraction from these roots deemed unsuitable for database construction (Appendix A), and therefore, RNA-Seq from waterlogged root was abandoned. The index-coded samples were clustered using TruSeq PE Cluster Kit v3-cBot-HS (Illumia) on a Cluster Generation System, following the manufacturer’s instructions. Once clusters had been generated, library preparations were sequenced on an Illumina Hiseq 2000 platform, with paired-end reads produced. Each sample yielded more than 6.03 Gb of clean data. Sequencing was completed by the Beijing Biomarker Biotechnology Company (Biomarker biotech, Beijing, China). The RNA-Seq data were obtained in three biological replicates.

### 4.5. Transcriptome Assembly and Differentially Expressed Genes (DEGs) Analysis

Transcriptome assembly was accomplished using the Trinity method [47] with min_kmer_cov set to 2 by default, with all other parameters set to default values. The expression abundance of gene was calculated using the Fragments Per Kilobase of transcript per Million mapped reads (FPKM) method [48]. To analyze DEGs, this was performed using the DESeq R package (1.10.1) between CK, WS30d, and WS30d-R10d. The resulting *p* values were adjusted using Benjamini and Hochberg’s approach to control the false discovery rate (FDR). Genes were considered differentially expressed when FDR was less than 0.01 and fold change (FC) was more than |1.5| between the treatments.

### 4.6. Annotation and Classification of DEGs

To annotate the functions of DEGs, we employed BLAST with an E-value ≦ 1 × 10^−10^-as the cutoff to align these sequences against the NCBI non-redundant database, COG, GO, and KEGG databases. This approach allowed us to assign putative functional annotations to each DEG, based on their sequence similarity to sequences in known databases.

### 4.7. Quantitative Real-Time PCR (qRT-PCR) Validation

To validate the RNA-seq results, we selected six genes from differentially expression genes for quantitative real-time PCR (qRT-PCR) analysis. Total RNA was extracted from the samples using the OmniPlant RNA Kit (CW2598S, Cwbiotech, Beijing, China) and reverse-transcribed using the TransScript All-in-One First-Strand cDNA Synthesis SuperMix for qPCR Kit (AT341-01, Transgen Biotech, Beijing, China), following the manufacturer’s protocol. The primer sequences used for qRT-PCR, including *β*-actin as an internal control, were listed in Appendix A. The qRT-PCR was performed using a Rotor-Gene Q real-time PCR system (Rotor-Gene, Qiagen, Germany). A total of 10 μL 2 X TransStart Top/TipGreen qPCR SuperMix (AQ141-02, Transgen Biotech, Beijing, China) was added to the reaction mixture, following the manufacturer’s instructions. The relative expression levels of the selected genes were normalized to the expression level of the *β*-actin gene. Cycle threshold values were then used to calculate expression levels using the 2^−ΔΔCt^ method [49].

### 4.8. Gas Chromatograph Coupled with a Time-of-Flight Mass Spectrometer (GC-TOF-MS) Analysis

The leaves from three individual seedlings were sampled and mixed as a duplicate, and the three duplicate samples were used for GC-MS detection. Metabolites were extracted from leave samples following the method described in [50], with some modifications. Approximately 20 mg of the sample was transferred into a 2 mL EP tubes and extracted with 500 μL of pre-cold extraction liquid (V_Methanol_:V_dH_2_O_ = 3:1). An aliquot of 10 μL of internal standard (adonitol) was added and mixed by vortexing for 30 s. The sample was then homogenized in a ball mill for 4 min at 35 Hz and subsequently treated with ultrasound for 5 min (incubated in ice water bath). The mixture was then centrifuged at 12,000 rpm at 4 °C for 15 min. The supernatant (100 μL) was transferred to a fresh 1.5 mL EP tube before being dried completely in a vacuum concentrator without heating. Next, the extracts were dissolved in 40 μL of methoxyamine hydrochloride (20 mg mL^−1^ in pyridine) and then incubated at 80 °C for 30 min. Finally, the samples were derivatized with 60 μL of N,O-bis(trimethylsilyl) trifluoroacetamide (BSTFA) reagent (with 1% TMCS, *v*/*v*) at 70 °C for 1.5 h. Quality control (QC) samples (a mixture of all samples to be analyzed) were also processed for detection.

Metabolite detection was conducted using an Agilent 7890 gas chromatography system coupled with a Pegasus HT time-of-flight mass spectrometer (Agilent Technologies, Santa Clara, CA, USA). The analysis was conducted in a randomized order after the addition of fatty acid methylesters (FAMEs). ADB-5MS capillary column coated with 5% diphenyl cross-linked with 95% dimethylpolysiloxane (30 m × 250 μm inner diameter, 0.25 μm film thickness; J&W Scientific, Folsom, CA, USA) was employed. An aliquot of 1 μL of the analyte was injected in splitless mode, while helium was used as a carrier gas. The front inlet purge flow was 3 mL min^−1^, and the gas flow rate through the column was 1 mL min^−1^. The initial temperature was held at 50 °C for 1 min, then raised to 310 °C at a rate of 10 °C min^−1^ and held at 310 °C for 8 min. The injection, transfer line, and ion source temperatures were maintained at 280 °C, 280 °C, and 250 °C, respectively. The energy was −70 eV in electron impact mode, and the mass spectrometry data were acquired in full-scan mode with an m/z range of 50–500 at a rate of 12.5 spectra per second after a solvent delay of 6.27 min.

### 4.9. Data Preprocessing and Compound Identification

Chroma TOF 4.3X software and the LECO-Fiehn Rtx5 database (LECO Corporation, Benton Harbor, MI, USA) were used for raw peak extraction, database line filtering and calibration, peak alignment, deconvolution analysis, peak identification, and integration of peak area. Both mass spectrum match and retention index match were considered during metabolite identification. Additionally, peaks detected in less than 50% of the QC samples or for which relative standard deviation (RSD) greater than 30% in the QC samples were removed [50,51]. Metabolites separated by GC-TOF-MS were identified using LECO ChromaTOF 4.3X software and the LECO/Fiehn Rtx5 metabolite mass spectral library by matching the mass spectrum and retention index.

### 4.10. Data Analysis

A three-dimensional data matrix was constructed, consisting of metabolite names (tentatively identified by GC-TOF-MS), sample information (three biological replicates of each treatment), and raw abundance (peak area for each tentatively identified metabolite) This matrix was uploaded to MetaboAnalyst 5.0 and analyzed in accordance with the provided instructions. Raw data were normalized using three categories of normalization: none, log transformation, and auto scaling. Multivariate analyses between CK, WS30d, and WS30d-R10d groups were performed using MetaboAnalyst 5.0, including orthogonal projections to latent structures-discriminant analysis (OPLS-DA) and principal component analysis (PCA). Moreover, the value of the variable importance in the projection (VIP) of the first principal component in OPLS-DA analysis was determined. Metabolites with VIP > 1 and *p* < 0.05 (student *t*-test) were considered significantly changed between groups.

### 4.11. Statistical Analysis

The data were presented as mean ± standard deviation (SD) and analyzed using either a one-way analysis of variance (ANOVA) or SNK *t*-test. Analysis was carried out in at least three replicates, with a *p* value < 0.05 considered statistically significant. The statistical analysis was performed using SPSS 18.0 for Windows. The difference test of gene expression among the three samples was performed using the R version 4.2.1 vegan package.

## 5. Conclusions

We utilized *R. delavayi* seedlings as a system to elucidate the response mechanism of alpine *Rhododendron* to waterlogging stress. Waterlogging stress was observed to reduce CO_2_ assimilation rate, stomatal conductance, transpiration rate, and maximum photochemical efficiency of PSII in the leave. The stress generated excessive H_2_O_2_, leading to oxidative stress. Additionally, waterlogging stress negatively affected lignin and cuticle biosynthesis, leading to leaf wilting, decreased carbohydrate transportation, and impeded sugar transport from leaves to roots through phloem, which ultimately results in sugar accumulation in the stressed leaves (Figure 8). Taken together, waterlogging stress has an irreversible effect on the *R. delavayi* seedlings in both physiological and molecular aspects. Regrettably, our study only shed light on the response mechanism of leaves, prompting further studies to develop into the stress response mechanism of young *R. delavayi* roots.

## Figures and Tables

**Figure 1 ijms-24-10509-f001:**
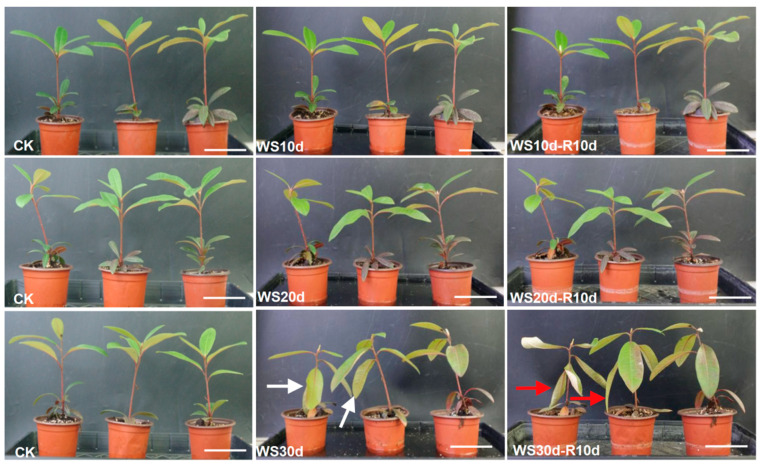
The leaf phenotype of *R. delavayi* after waterlogging stress and post-waterlogging recovery. Control (CK), waterlogging stress for 10 days (WS10d), 20 days (WS20d), and 30 days (WS30d) were indicated. Post-waterlogging recovery for 10 days was indicated by WS10d-R10d, WS20d-R10d, WS30d-R10d. Wilting of leaves after WS30d and WS30d-R10d was represented by white and red arrow, respectively. Bar = 5.8 cm.

**Figure 2 ijms-24-10509-f002:**
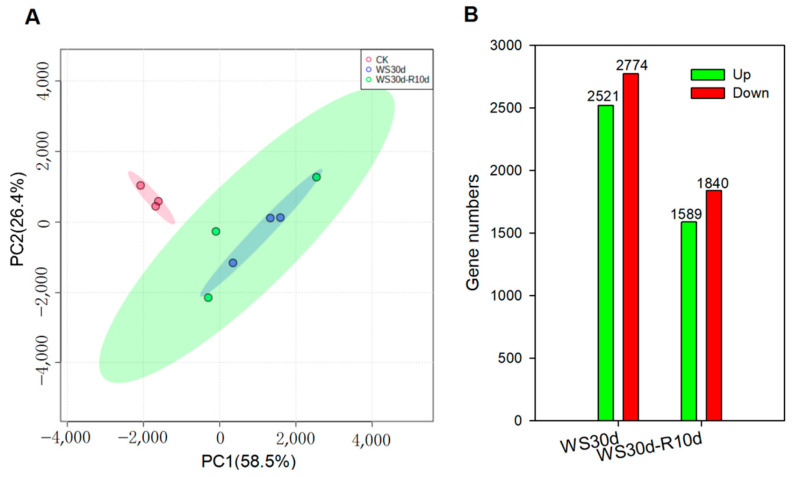
The principal component analysis (PCA) and differentially expressed genes (DEGs). (**A**) PCA of all detected genes; (**B**) DEGs in WS30d and WS30d-R10d leaves in *R. delavayi* compared to CK.

**Figure 3 ijms-24-10509-f003:**
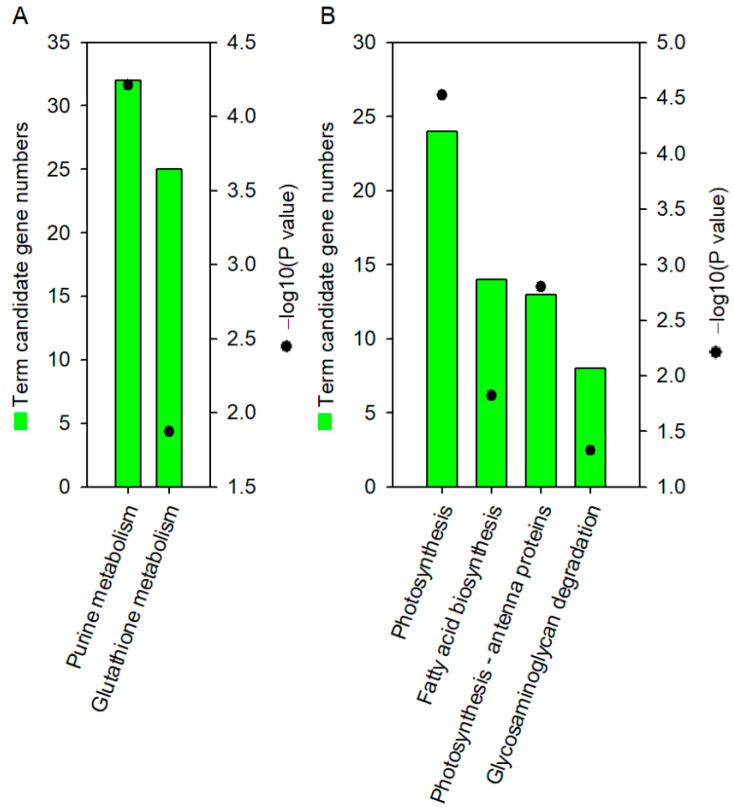
KEGG pathways of DEGs in WS30d leaves response to waterlogging stress. (**A**) up-regulation DEGs enriched in KEGG; (**B**) down-regulation DEGs enriched in KEGG.

**Figure 4 ijms-24-10509-f004:**
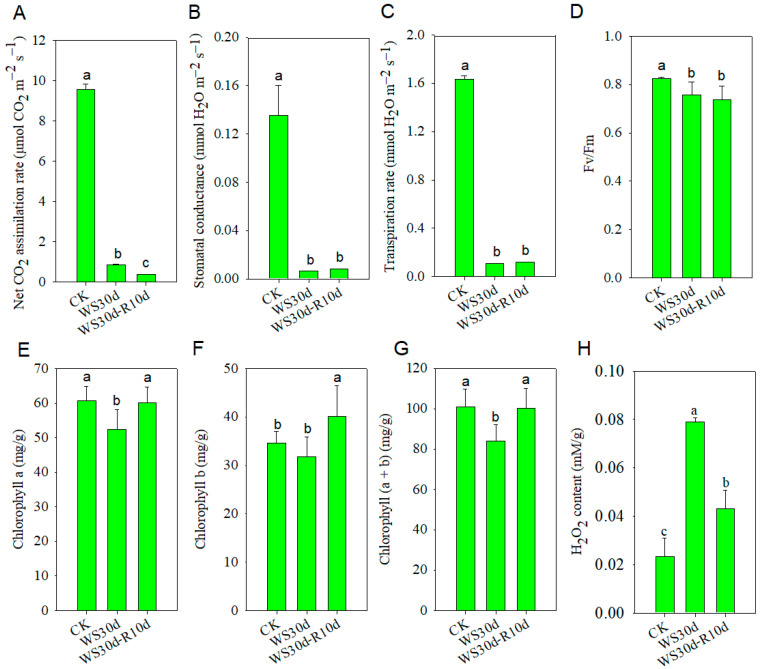
Photosynthesis, chlorophyll content and H_2_O_2_ content in CK, WS30d and WS30d-R10d leaves. (**A**) Net CO_2_ assimilation rate; (**B**) stomatal conductance; (**C**) transpiration rate; (**D**) maximum photochemical efficiency; (**E**) Chlorophyll a content; (**F**) Chlorophyll b content; (**G**) Chlorophyll (a + b); and (**H**) H_2_O_2_ content. Different lowercase letters represented significant differences between treated groups and CK (*p* ˂ 0.05).

**Figure 5 ijms-24-10509-f005:**
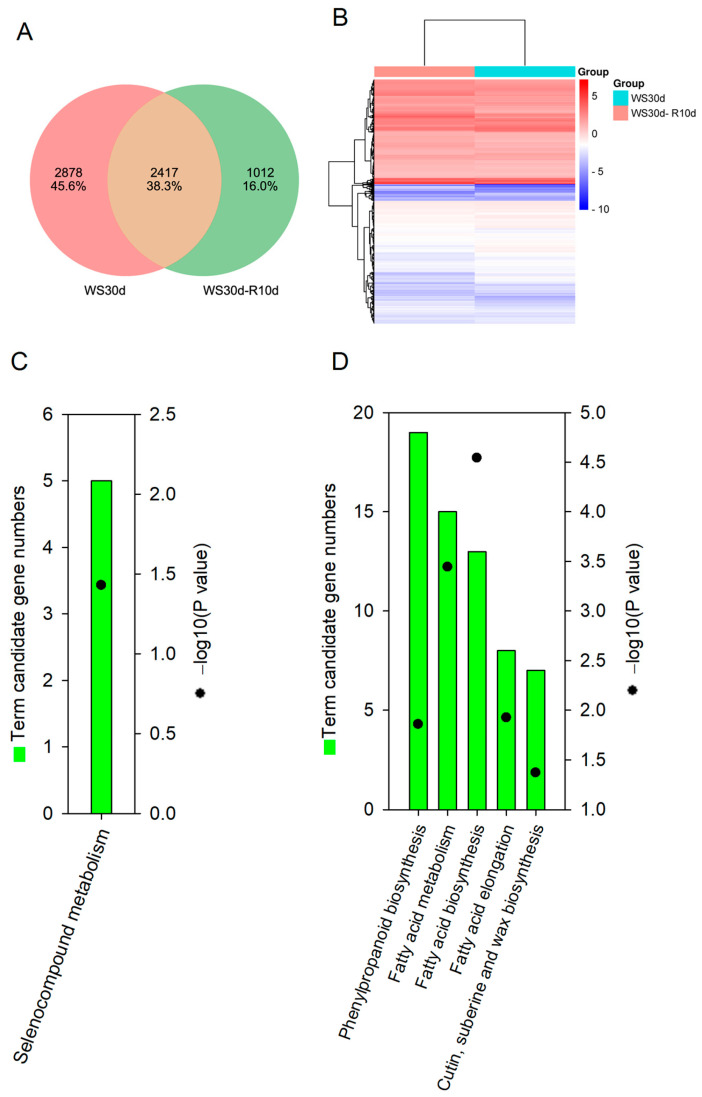
KEGG pathways of common DEGs in *R. delavayi* leaves between WS30d and WS30d-R10d. (**A**) Venn diagram of DEGs with annotations identified by RNA-Seq between WS30d and WS30d-R10d; (**B**) heat maps of 2417 common DEGs in (**A**); (**C**) up-regulation DEGs enriched in KEGG; (**D**) down-regulation DEGs enriched in KEGG.

**Figure 6 ijms-24-10509-f006:**
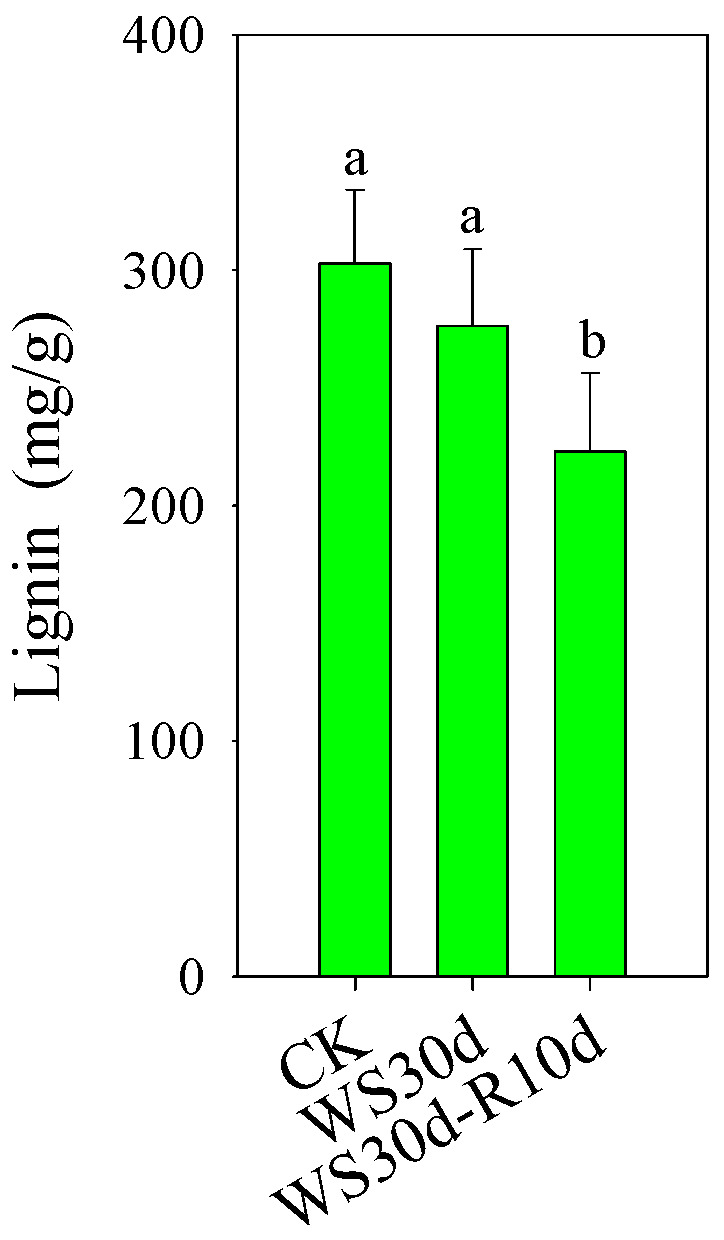
Lignin content in CK, WS30d and WS30d-R10d leaves. Different lowercase letters represented significant differences between treated groups and CK (*p* ˂ 0.05).

**Figure 7 ijms-24-10509-f007:**
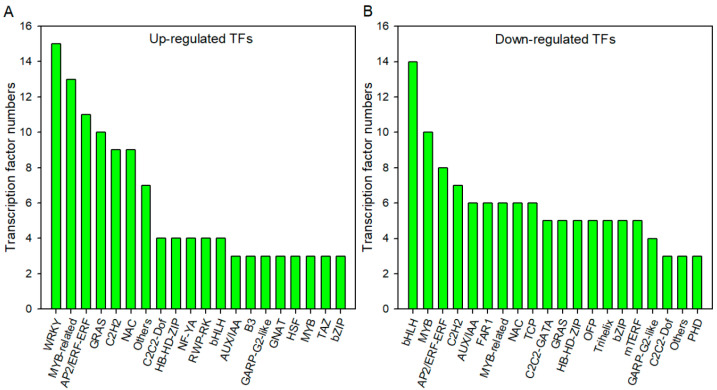
The predicting transcription factors (TFs) encoded by DEGs in WS30d leaves. (**A**) the TFs encoded by up-regulated DEGs; (**B**) the TFs encoded by down-regulated DEGs.

**Figure 8 ijms-24-10509-f008:**
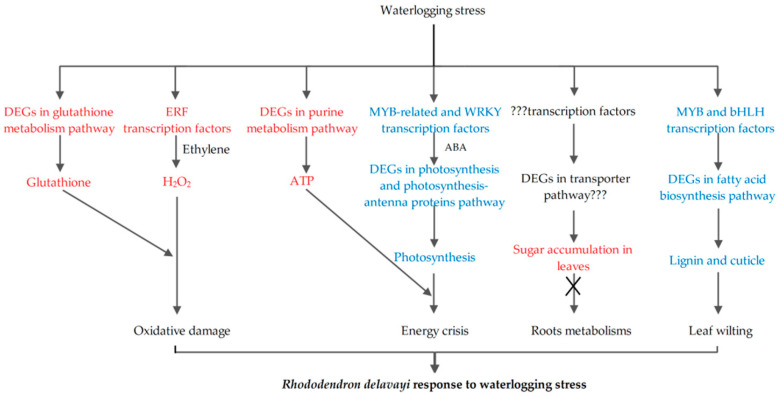
Simplified scheme of the response in *R. delavayi* leaves to waterlogging stress. Genes and metabolites indicated in blue are usually down-regulated and decreased in abundance, respectively. Genes and metabolites indicated in red are usually up-regulated and increased in abundance, respectively. ??? represents unknown transcription factors and pathways.

**Table 1 ijms-24-10509-t001:** DEGs of WS30d versus CK enriched in KEGG pathways (*p* ˂ 0.05).

#ID	FPKM		FDR	log_2_FC	KEGG_Annotation
	CK	WS30d			
Purine Metabolism					
c61670.graph_c0	24	121.11	5.95 × 10^−8^	2.32	3′-phosphoadenosine 5′-phosphosulfate synthase
c68596.graph_c0	21.65	222.9	1.81 × 10^−10^	2.88	3′-phosphoadenosine 5′-phosphosulfate synthase
c48065.graph_c0	24.05	66.58	0.009568452	1.19	adenylate kinase
c55902.graph_c0	5.9	17.61	0.002497335	1.31	adenylate kinase
c61967.graph_c0	31.45	45.22	0.001423826	1.30	amidophosphoribosyltransferase
c54389.graph_c0	0.69	2.94	5.56 × 10^−8^	1.77	AMP deaminase
c56269.graph_c0	5.46	9.79	0.002969771	0.68	DNA polymerase delta subunit 3
c61104.graph_c0	3.61	7.69	3.14 × 10^−5^	0.94	DNA polymerase I
c67065.graph_c1	11.34	23.74	2.70 × 10^−5^	0.72	DNA polymerase I
c61560.graph_c1	7.9	16.7	0.000143442	1.37	DNA primase small subunit
c66963.graph_c1	7.71	15.06	0.002704535	0.80	DNA-directed RNA polymerase I subunit RPA1
c66139.graph_c0	6.19	15.02	4.30 × 10^−5^	1.08	DNA-directed RNA polymerase I subunit RPA2
c65110.graph_c0	0.12	1.13	0.00607105	2.12	DNA-directed RNA polymerase II subunit RPB1
c54215.graph_c1	4.71	10.21	0.001854865	0.87	DNA-directed RNA polymerase II subunit RPB4
c55954.graph_c1	3.04	5.27	0.003121169	0.73	DNA-directed RNA polymerase III subunit RPC1
c63522.graph_c0	14.72	29.28	1.94 × 10^−6^	0.84	DNA-directed RNA polymerase III subunit RPC2
c61042.graph_c0	0.61	2.01	0.006880565	1.56	DNA-directed RNA polymerase subunit
c61639.graph_c0	32.03	62.06	7.63 × 10^−5^	0.80	DNA-directed RNA polymerase subunit
c44354.graph_c0	3	10.15	1.84 × 10^−5^	1.53	DNA-directed RNA polymerases I and III subunit RPAC1
c66477.graph_c0	40.5	123.45	2.42 × 10^−9^	1.43	hydroxyisourate hydrolase
c61783.graph_c1	53.95	124.98	0.004522204	1.06	nucleoside-diphosphate kinase
c56191.graph_c0	15.05	37.97	2.23 × 10^−5^	1.14	phosphoglucomutase
c66317.graph_c0	0.56	2.54	5.01 × 10^−8^	1.88	polyribonucleotide nucleotidyltransferase
c46598.graph_c0	12.84	30.73	0.001705575	1.06	pyruvate kinase
c56447.graph_c0	11.28	30.88	0.000148805	1.24	pyruvate kinase
c46598.graph_c1	13.17	35.15	0.005231922	1.16	pyruvate kinase
c67771.graph_c1	20.81	59.57	0.001272097	1.27	pyruvate kinase
c65971.graph_c0	3.1	8.33	1.43 × 10^−7^	1.27	ribonucleoside-diphosphate reductase subunit M1
c65971.graph_c2	3.17	7.96	1.74 × 10^−5^	1.16	ribonucleoside-diphosphate reductase subunit M1
c51808.graph_c0	2.91	8.08	1.11 × 10^−6^	1.32	ribonucleoside-diphosphate reductase subunit M2
c59718.graph_c0	2.19	7.62	1.01 × 10^−6^	1.60	ribose-phosphate pyrophosphokinase
c65143.graph_c1	35.14	88.48	0.001053369	1.18	urate oxidase
Glutathione metabolism					
c58797.graph_c0	28.29	59.89	0.000787329	0.82	glutamate-cysteine ligase
c49129.graph_c0	17.02	64.83	0.000110947	1.66	glutathione peroxidase
c56863.graph_c0	45.09	111.71	3.73 × 10^−13^	1.11	glutathione reductase
c59612.graph_c0	94.14	241.87	0.002044808	1.18	glutathione reductase
c64812.graph_c0	18.45	130.44	2.77 × 10^−5^	1.92	glutathione S-transferase
c59164.graph_c0	1.5	11.4	3.70 × 10^−6^	2.53	glutathione S-transferase
c63277.graph_c1	25.21	729.15	3.79 × 10^−9^	3.91	glutathione S-transferase
c64156.graph_c0	11.59	23.7	0.000578537	0.95	glutathione S-transferase
c60034.graph_c0	1.33	89.58	0.000390662	3.12	glutathione S-transferase
c47361.graph_c0	2.25	13.34	0.000218054	2.10	glutathione S-transferase
c61754.graph_c0	1.38	3.79	0.006001931	1.04	glutathione S-transferase
c51875.graph_c0	102.67	346.18	0.00021695	1.49	glutathione S-transferase
c50282.graph_c0	18.24	139.8	9.15 × 10^−7^	2.53	glutathione S-transferase
c67680.graph_c0	149.26	673.04	0.000191146	1.90	glutathione S-transferase
c65078.graph_c0	19.02	45.97	2.39 × 10^−5^	1.14	glutathione S-transferase
c60993.graph_c0	4.01	18.46	3.33 × 10^−6^	2.04	glutathione S-transferase
c65367.graph_c0	37.95	463.28	1.92 × 10^−11^	3.32	glutathione S-transferase
c50305.graph_c0	1.07	8.46	0.002469129	2.14	glutathione S-transferase
c64290.graph_c0	9.52	84.97	2.20 × 10^−6^	2.72	glutathione S-transferase
c52719.graph_c0	5.53	21.94	6.37 × 10^−5^	1.82	glutathione synthase
c62411.graph_c0	5.21	15.07	7.17 × 10^−8^	1.36	isocitrate dehydrogenase
c55430.graph_c0	287.94	520.51	0.00599229	0.64	L-ascorbate peroxidase
c65971.graph_c0	3.1	8.33	1.43 × 10^−7^	1.27	ribonucleoside-diphosphate reductase subunit M1
c65971.graph_c2	3.17	7.96	1.74 × 10^−5^	1.16	ribonucleoside-diphosphate reductase subunit M1
c51808.graph_c0	2.91	8.08	1.11 × 10^−6^	1.32	ribonucleoside-diphosphate reductase subunit M2
Photosynthesis					
c68574.graph_c2	46.26	13.35	5.24 × 10^−8^	−2.1001	cytochrome b6/f
c50907.graph_c1	158.23	86.54	1.12 × 10^−6^	−1.078	ferredoxin
c44849.graph_c0	40.66	26.51	0.000278672	−0.7568	ferredoxin-NADP+ reductase
c63271.graph_c3	62.05	18.73	5.44 × 10^−11^	−1.7936	F-type H+-transporting ATPase subunit a
c57998.graph_c0	67.2	20.66	1.48 × 10^−12^	−2.0903	F-type H+-transporting ATPase subunit alpha
c61006.graph_c0	469.97	258.81	0.00101548	−0.9944	F-type H+-transporting ATPase subunit delta
c55205.graph_c0	1.32	0.29	0.003579127	−1.9289	F-type H+-transporting ATPase subunit epsilon
c67974.graph_c4	112.42	21.02	3.21 × 10^−8^	−2.3965	photosystem I P700 chlorophyll a apoprotein A1
c66205.graph_c2	1099.4	676.68	0.005253358	−0.8404	photosystem I subunit II
c44286.graph_c0	1447.6	631.02	0.000768555	−1.3079	photosystem I subunit IV
c43761.graph_c0	1209.2	551.7	0.000426669	−1.2402	photosystem I subunit PsaN
c50596.graph_c0	603.5	317.38	5.96 × 10^−5^	−1.0638	photosystem I subunit V
c47764.graph_c0	2073.3	1148.2	0.001696486	−0.9827	photosystem I subunit VI
c50541.graph_c0	1059.1	428.24	0.00013698	−1.3976	photosystem I subunit X
c46857.graph_c0	1711	892.21	0.000848878	−1.0656	photosystem I subunit XI
c63847.graph_c0	28.52	4.57	3.13 × 10^−15^	−2.6784	photosystem II CP43 chlorophyll apoprotein
c63896.graph_c0	32.78	14.67	7.23 × 10^−7^	−1.3887	photosystem II CP43 chlorophyll apoprotein
c55414.graph_c0	37.83	8.26	1.30 × 10^−10^	−2.3996	photosystem II CP47 chlorophyll apoprotein
c48666.graph_c0	42.45	22.47	9.08 × 10^−5^	−1.0589	photosystem II oxygen-evolving enhancer protein 2
c56596.graph_c0	1509.8	767.15	0.000796919	−1.1323	photosystem II oxygen-evolving enhancer protein 3
c62948.graph_c3	391.53	79.36	9.84 × 10^−12^	−2.3642	photosystem II P680 reaction center D1 protein
c59404.graph_c0	46.69	10.13	5.63 × 10^−11^	−2.2111	photosystem II P680 reaction center D2 protein
c53605.graph_c0	1317.5	669.41	2.04 × 10^−6^	−1.1168	photosystem II PsbW protein chloroplastic-like
c65097.graph_c0	49.88	25.08	8.91 × 10^−5^	−1.1178	photosystem II PsbY protein
Fatty acid biosynthesis					
c67210.graph_c2	5.34	3.78	0.005595171	−0.8695	3-oxoacyl-[acyl-carrier-protein] synthase II (FabF)
c57955.graph_c0	44.1	21.27	1.60 × 10^−8^	−1.0614	3-oxoacyl-[acyl-carrier-protein] synthase II (FabF)
c58354.graph_c0	12.57	0.62	2.47 × 10^−12^	−3.9695	3-oxoacyl-[acyl-carrier-protein] synthase II (FabF)
c62774.graph_c0	32.83	20.68	5.66 × 10^−5^	−0.7974	3-oxoacyl-[acyl-carrier-protein] synthase III (FabH)
c53077.graph_c0	29.14	9.74	2.39 × 10^−11^	−1.6865	acetyl-CoA carboxylase biotin carboxyl carrier protein
c67708.graph_c1	60.53	43.04	0.001805159	−0.638	acetyl-CoA carboxylase carboxyl transferase subunit alpha
c46335.graph_c0	50.52	31.1	4.92 × 10^−5^	−0.8391	acetyl-CoA carboxylase, biotin carboxylase subunit
c46455.graph_c0	31.28	20.51	0.002644038	−0.7592	enoyl-[acyl-carrier protein] reductase I (FabI)
c50059.graph_c0	36.92	11.05	2.67 × 10^−9^	−1.8325	enoyl-[acyl-carrier protein] reductase I (FabI)
c60574.graph_c0	55.26	7.04	5.88 × 10^−24^	−2.7932	fatty acyl-ACP thioesterase A
c64735.graph_c0	242.08	108.24	3.08 × 10^−12^	−1.3285	fatty acyl-ACP thioesterase B
c66291.graph_c1	252.71	32.02	6.43 × 10^−28^	−3.0528	long-chain acyl-CoA synthetase
c59094.graph_c0	71.72	20.79	6.86 × 10^−23^	−2.2218	long-chain acyl-CoA synthetase
c56069.graph_c0	29.9	17.27	1.13 × 10^−5^	−1.0826	S-malonyltransferase
Photosynthesis—antenna proteins					
c51141.graph_c0	1625.3	463.16	1.29 × 10^−5^	−1.838	light-harvesting complex I chlorophyll a/b binding protein 1
c62753.graph_c1	949.2	442.35	0.000170897	−1.2306	light-harvesting complex I chlorophyll a/b binding protein 2
c65754.graph_c0	3672.8	2120.8	0.008139475	−0.9203	light-harvesting complex I chlorophyll a/b binding protein 3
c41195.graph_c0	2546.7	785.55	2.25 × 10^−5^	−1.7575	light-harvesting complex I chlorophyll a/b binding protein 4
c65915.graph_c0	66.33	39.48	4.72 × 10^−5^	−0.8903	light-harvesting complex I chlorophyll a/b binding protein 5
c32187.graph_c0	23.74	0.61	2.53 × 10^−8^	−4.0809	light-harvesting complex II chlorophyll a/b binding protein 1
c63116.graph_c0	18325	7434.7	3.70 × 10^−6^	−1.5994	light-harvesting complex II chlorophyll a/b binding protein 1
c68500.graph_c1	294.76	87.78	2.92 × 10^−6^	−1.8158	light-harvesting complex II chlorophyll a/b binding protein 2
c49775.graph_c0	30.49	8.1	6.84 × 10^−5^	−1.9016	light-harvesting complex II chlorophyll a/b binding protein 2
c52982.graph_c0	1074.5	316.12	6.09 × 10^−5^	−1.7878	light-harvesting complex II chlorophyll a/b binding protein 3
c56361.graph_c0	1720.6	769.22	0.000795542	−1.2611	light-harvesting complex II chlorophyll a/b binding protein 4
c65425.graph_c0	1478	735.17	0.000311123	−1.1253	light-harvesting complex II chlorophyll a/b binding protein 5
c52323.graph_c0	1566.1	601.7	0.000807719	−1.4493	light-harvesting complex II chlorophyll a/b binding protein 6
Glycosaminoglycan degradation					
c38948.graph_c0	10.24	3.43	9.09 × 10^−6^	−1.6616	heparanase
c38920.graph_c0	36.54	25.82	8.92 × 10^−6^	−0.6415	heparanase
c65686.graph_c0	15.85	6.55	1.99 × 10^−9^	−1.4827	heparanase
c49429.graph_c0	2.2	0.05	4.01 × 10^−11^	−4.5862	heparanase
c68546.graph_c0	28.62	14.23	1.60 × 10^−12^	−1.2804	heparanase
c59153.graph_c0	66.93	39.86	4.05 × 10^−6^	−0.9714	hexosaminidase
c64000.graph_c0	40.48	14.83	2.73 × 10^−5^	−1.5638	hexosaminidase

**Table 2 ijms-24-10509-t002:** Differential metabolites in *R. delavayi* leaves under waterlogging stress for 30 days (WS30d).

Class	Putative Metabolites	CK	WS30d	VIP	*p* Value	FC	Log_2_FC
Sugars	Glucose	1.020305382	1.885810307	1.414	0.026	1.848	0.886
	Sedoheptulose	0.039939754	0.384491416	1.674	0.026	9.627	3.267
	Galactose	0.242448413	0.678832861	1.637	0.004	2.800	1.485
	Sucrose	0.513948256	1.015253352	1.634	0.001	1.975	0.982
	Lyxose	0.01002245	0.014270754	1.459	0.034	1.424	0.510
	Galactonic acid	0.027194517	0.054436394	1.452	0.039	2.002	1.001
	N-Acetyl-beta-D-mannosamine	0.008378467	0.016720616	1.480	0.023	1.996	0.997
Organic acids	L-Malic acid	0.506445001	1.243612412	1.505	0.044	2.456	1.296
	Citric acid	0.145507649	0.462412497	1.373	0.019	3.178	1.668
	Glucoheptonic acid	0.180491598	0.395956908	1.627	0.001	2.194	1.133
Esters	Gluconic lactone	0.22506794	0.492767481	1.616	0.002	2.189	1.131
	beta-Mannosylglycerate	0.028496106	0.056197656	1.354	0.049	1.972	0.980
	L-Gulonolactone	0.005848757	0.050757515	1.277	0.006	8.678	3.117
Fatty acid	Linolenic acid	0.087711244	0.205478671	1.505	0.017	2.343	1.228
Alcohols	Diglycerol	0.055957005	0.12707336	1.501	0.025	2.271	1.183
	Sorbitol	0.008703194	0.024159015	1.537	0.020	2.776	1.473
Flavonoids	Arbutin	12.28242997	24.66825634	1.521	0.008	2.008	1.006
	Phenyl beta-D-glucopyranoside	0.007043534	0.018592753	1.623	0.011	2.640	1.400
Others	p-Coumaric acid	0.104651639	0.363258482	1.620	0.001	3.471	1.795
	Glycocyamine	0.005322693	0.028323088	1.261	0.016	5.321	2.412
	Glutathione	0.006044401	0.011154309	1.370	0.048	1.845	0.884

## Data Availability

The data presented in this study are available on request from the corresponding author.

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
