# Peer review of "Transcriptomic, Physiological, and Metabolomic Response of an Alpine Plant, Rhododendron delavayi, to Waterlogging Stress and Post-Waterlogging Recovery"

_ijms, 2023, doi:10.3390/ijms241310509_

Round 1
Reviewer 1 Report
The manuscript entitled “Transcriptomic, Physiological and Metabolomic Response of an Alpine Plant, Rhododendron delavayi, to Waterlogging Stress and Post-waterlogging Recovery” by Zhang et al. aimed to explore the mechanisms in response to waterlogging stress in woody Rhododendron trees. The authors have indicated that R. delavayi is a waterlogging sensitive species base on their transcriptomic, physiological and data analysis. The experimental approach could have been better to analyse the waterlogging stress in Rhododendron. They have also presented some of the data in a very general way without critically analysing it.
I have the following major comments for the authors:
1. The Introduction needs major revision They need to explain the molecular mechanism of waterlogging stress including metabolic, hormonal and transcriptional regulatory pathways giving example of other plants. They have written a very general introduction.
2. The major concern is the control data. They do not have control data for 10, 20 and 30 days old seedlings without waterlogging stress. They have indicated the initial day as control data. It would have been better if they would have compared seedlings with/without waterlogging stress.
3. The authors have mainly compared the data between Ck, WS30d and WS30d-R10d. What about WS10d and WS10d-R10d, and WS20d and WS20d-R10d? Waterlogging stress for 30 days is quite long. They could be tolerant to waterlogging stress for shorter days. If you have the data you can compare between 10, 20 and 30 days.
4. Please provide the differential gene expression data in a supplementary excel file.
5. Discussion also needs major revision mainly on transcription factors (TFs) and hormonal signalling part. The authors have just mentioned regarding the up/down regulation of genes. However, they have not discussed the role of those transcription factors in response to waterlogging stress. The results show up regulation of a number of TFs. There is also no information on any phytohormones. Phytohormones play a major role during any stress condition.
6. Figure 8 can be improved based on their revised analysis.
7. Please check the language throughout the manuscript.
The authors need to refine the language.
Reviewer 2 Report
Dear Authors,
Congratulations for this work. It was notworthy inded, to try this on a plant as Rhododendron.
I have some concerns: There are lots of typographical errors. I have highlighted a few.
Line 102: It should be total
Line 117: Font size not same
Line 189: Its Venn not venn
Line 262: space problem
1. Can you mention a list of genes in the conclusion which can be used to combat waterlogging stress in other transgenics.
2. Please highlight in the figures the leaves which are showing water logging effect. You did show in only one.
3.Please confirm how many repeats/samples were taken how GC-MS analysis? How was the internal control used in the calculation?
4. Can we get the p values mentioned for all the a, b used as stats throughout the paper?
5. Figure 5b: Can the heatmap be made a little bit readable? The DEGs can be made look better.
6. Was there any physiological estimates for parameters as lignin content? glucose, ROS, etc to validate the proofs you got from GCMS?
7. The discussion is too long and very factual kind. I would ask the authors to prove some of the genes. Realtime PCR or any other method of quantitative analysis.
8. Is there a possibility that this study can be done with roots as in waterlogging roots are the first source of contact. Can these plants/seedlings be grown in hydrophonic? You can discuss this in the conclusion.
Serious typo errors, font size issue, spacing problems.
Round 2
Reviewer 1 Report
Thank you authors for revising the manuscript.
I have few minor comments for the authors:
1. Please revise the following lines as transcription factors are hormone responsive.
a. They regulate the transcription of genes in hormone synthesis pathway and secondary metabolite biosynthesis pathway, thereby regulating the response to waterlogging stress [78-9]. (Line 59-61)
b. ERF transcription factors regulate the expression of genes in ethylene biosynthesis [32, 28], which is involved in various physiological processes under waterlogging stress, including shoot growth, adventitious root elongation, and ROS accumulation [34]. (Line 350-352)
2, Please check the language throughout the manuscript.
Please refine the language throughout the manuscript.
Reviewer 2 Report
Dear Authors,
Congratulations on presenting such a nice work.
All the comments have been addressed to the best.
Author Response
Thank you very much for your carefully reading to our manuscript again.